# Adapting a medication adherence app for adolescents and young adults in Makurdi, Benue State: The Lu Dedoo Project

Olaposi Joseph Olatoregun[1,2]⊛*, Lisa Hightow-Weidman[3]⊛, Marta Mulawa[4]⊛, Lauren Jennings[2]‡, Jay Osi Samuels[1]‡, Prosper Okonkwo[1]‡, Catherine Orrell[2]⊛

1 PHIS3 Project, APIN Public Health Initiatives, Abuja, Nigeria, 2 Desmond Tutu HIV Centre, University of Cape Town, Cape Town, South Africa, 3 College of Nursing, Florida State University, Tallahassee, Florida, United States of America, 4 School of Nursing, Duke University, Durham, North Carolina, United States of America

⊛ These authors contributed equally to this work.
‡ LJ, JO and PO also contributed equally to this work.
* olaposij@yahoo.co.uk

## Abstract

### Introduction

Adolescents and young adults living with HIV (AYALHIV) face persistent challenges with antiretroviral therapy (ART) adherence, a key determinant of viral suppression and long-term health outcomes. The *Lu Dedoo Project* sought to adapt the existing (Masakhane Siphucule Impilo Yethu (MASI) adherence app, originally developed for South Africa, for AYALHIV in Benue State, Nigeria. Guided by a human-centered design (HCD) framework, the project engaged end users to ensure that the app was contextually relevant, user-friendly, and responsive to local needs prior to pilot testing.

### Methods

Beta testing was conducted where purposively selected AYALHIV used MASI in its existing form for one month. Three subsequent FGDs were held among beta testers, involving a total of 22 participants: one mixed-gender adolescent group aged 15–19 years (n = 6), one male young adult group aged 20–24 years (n = 8), and one female young adult group aged 20–24 years (n = 8). FGDs were analysed using thematic analysis in NVivo® v15. Transcripts were analysed thematically to identify user experiences and desired modifications. Identified features were then prioritised using the MoSCoW method ("Must have," "Should have," "Could have," "Won't have") to guide development decisions and ensure feasible, user-driven adaptation.

**Data availability statement:** The minimal data set and accompanying analytic materials are available at Mendeley Data via https://doi.org/10.17632/vwmmnhgsz6.1.

**Funding:** This work was supported by the European & Developing Countries Clinical Trials Partnership 2 (EDCTP2) programme under grant number EDCTP TMA2019SFP-2812, through the Desmond Tutu Health Foundation, South Africa. There was no additional external funding received for this study.

**Competing interests:** The authors have declared that no competing interests exist.

## Results

Participants found the app highly engaging and liked the games, quizzes, and the ability to earn badges. These features contributed to user retention and supported their medication adherence. A subset of participants noted the challenge of needing cellular data access to use the app. "Must have" priorities included offline access to key features, simplified navigation, and customizable medication reminders. "Should have" and "Could have" priorities included adding more engaging media content and enhancing personalised reminders to better meet user needs.

## Conclusions

Applying an HCD framework, complemented by MoSCoW prioritisation, provided a structured and participatory pathway for adapting an mHealth adherence app to the Nigerian context. The process ensured that core user needs guided technical decisions and that limited development resources were directed toward high-impact features. The adapted *Lu Dedoo* app reflects local preferences for functionality, language, and offline usability, and represents a promising, context-specific tool for improving ART adherence among AYALHIV.

## Introduction

Adolescents and young adults living with HIV (AYALHIV) face persistent challenges with adherence to antiretroviral therapy (ART), resulting in increased risks of virologic failure, drug resistance, and poor health outcomes [1]. These challenges are especially pronounced in sub-Saharan Africa, where structural barriers such as stigma, limited psychosocial support, socioeconomic barriers such as low income, employment status, and health systems factors such as stockout of medical supplies and unavailability of trained healthcare personnel, compound individual and social vulnerabilities [1–4].

In Nigeria, Benue State is noted for having one of the highest HIV prevalence rates in the country at 4.9% [5]. Adolescents and young adults in this state constitute a priority demographic for intervention, as they represent approximately 60% of the population and exhibit the highest HIV prevalence. [5,6].

Mobile health (mHealth) technologies have emerged as promising tools for promoting ART adherence, especially among youth who increasingly rely on smartphones for communication and daily tasks [7]. The proliferation of affordable smartphones and the expansion of mobile internet coverage across sub-Saharan Africa have created unprecedented opportunities to deliver health interventions directly into the hands of end users. In the context of HIV care, mHealth applications offer a versatile platform to address the multifactorial barriers to adherence by combining behavioural support with real-time monitoring and personalized feedback. Evidence from populations in low- and middle-income countries demonstrates that well-designed mHealth apps can improve medication adherence by incorporating

features such as medication reminders, symptom tracking, peer support networks, educational content, gamification elements, and real-time adherence tracking linked to clinic databases [8,9].

In addition, mHealth applications have shown feasibility and acceptability in similar populations by offering features such as medication reminders, peer support, gamification, and real-time adherence tracking [10]. In particular, co-designing digital interventions with input from the end-users, adolescents and healthcare workers, enhances app relevance, usability, and uptake, especially in low-resource settings [11]. This participatory approach, often described as human-centered design (HCD), ensures that apps are not only technically functional but also socially acceptable, linguistically appropriate, and responsive to contextual constraints such as internet costs, stigma, and privacy concerns [12].

The Masakhane Siphucule Impilo Yethu (MASI) application, originally developed in South Africa and adapted from the HealthMpowerment app and platform, is a smartphone app designed to support ART adherence in AYALHIV.

The MASI app includes interactive features such as adherence reminders, health education content, discussion forums, and personal tracking tools. Early evaluations of MASI suggest it is feasible, culturally appropriate, and potentially impactful in improving ART adherence among South African adolescents [13–15].

In the present study, we describe the initial development and contextual adaptation of the MASI app for use among AYALHIV in Benue State, Nigeria. This phase of the project involved a series of focus group discussions (FGDs) with AYALHIV. These findings directly informed the customization of the app to reflect the sociocultural and health system context of the target population. This user-centered, formative phase sets the foundation for the subsequent randomized controlled trial to evaluate the feasibility, acceptability, and effectiveness of the adapted app in improving ART adherence in this high-burden region.

## Materials and methods

### Study design

This study employed a modified HCD process focusing on formative qualitative input to inform the contextual adaptation of the MASI mobile app for AYALHIV in Benue State, Nigeria [12]. HCD emphasizes participatory processes in which end users actively contribute to the identification of needs and the co-creation of solutions. The formative phase of the project included a series of FGDs with AYALHIV aimed at identifying adherence barriers, assessing user experiences with an early version of the MASI app, and generating insights to guide app customization.

Following thematic analysis, and to ensure systematic prioritization of suggested features, we applied the Must have, Should have, Could have, and Won't have (MoSCoW) method which allowed us to balance user preferences with technical feasibility and resource constraints [16,17]. This dual framework ensured that the adaptation process remained both user-driven and practically implementable.

### Study setting

The study was conducted at three high-volume HIV treatment facilities in Makurdi Local Government Area, Benue State. These sites are supported by APIN Public Health Initiatives under the PEPFAR program and collectively provide ART to over 19,000 clients, including a significant adolescent and young adult population.

### Participant recruitment

Participants were recruited through consecutive recruitment at the three HIV treatment clinics in Benue State, Nigeria. Individuals were eligible to participate if they were aged 15–24 years at the time of recruitment (adolescents aged 15–19 years and young adults aged 20–24 years), were currently receiving ART at one of the participating sites, owned an Android smartphone, and demonstrated basic digital literacy. Demographic data were collected at the time of participation. To ensure confidentiality, participants were assigned anonymous pseudonyms for all transcripts and study records.

 

## Ethical considerations

Written informed consent was obtained from all adult participants, and assent was obtained from adolescents under 18 years, along with guardian consent. Confidentiality and data security were maintained throughout the study. Ethical approval for the study was obtained from the National Health Research Ethics Committee of Nigeria (NHREC) with approval number NHREC/01/01/2007–15/06/2024D.

## Pre-FGD app exposure

To support informed and meaningful participation in the FGDs, the existing version of the MASI app, originally developed for adolescents living with HIV in South Africa, was installed on participants' Android smartphones approximately one month prior to their scheduled FGD. The recruitment and installation commenced on 4th September 2023 and ended on 2nd October 2023.

At the time of installation, participants received a brief orientation on how to access and navigate the app. Over the next four weeks, participants were encouraged to explore all app features independently at their own pace. No formal training, usage goals, or structured engagement requirements were imposed during this period. This approach allowed users to develop familiarity with the app's content and functionality in a naturalistic manner, thereby enabling more grounded feedback during the FGDs.

## Focus group design and implementation

Three FGDs were conducted to gather feedback on the contextual adaptation of the MASI app. In total, 22 participants took part in the discussions. Groups were stratified by age and gender to promote comfort and peer comparability: (1) one mixed-gender adolescent group aged 15–19 years (n = 6), (2) one male young adult group aged 20–24 years (n = 8), and (3) one female young adult group aged 20–24 years (n = 8). Participants were recruited consecutively during routine clinic visits at the participating study sites.

All FGDs were conducted in private rooms within the participating health facilities to ensure confidentiality and participant comfort. Discussions were facilitated using FGD interview guides developed for this study. The discussion guide was developed in English and administered in English; no translation was required. To support participation, each individual received a transport stipend of 5,000 naira. Lunch was also provided during the sessions.

## Data management

Participant information, including names and mobile phone numbers, was stored in a separate, password-protected file accessible only to designated study personnel. This information was used solely for scheduling FGDs and administering participant compensation. All physical data (e.g., printed transcripts or handwritten notes) were stored in locked file cabinets at the study coordination offices and accessed only when in use. Digital data, including audio recordings and typed transcripts, were stored on secure, limited-access network drives maintained by the lead research institution. To safeguard confidentiality, participants were assigned pseudonyms in all transcripts and written materials. Participants were also reminded not to disclose any identifying information or details discussed during the FGDs to individuals outside the group.

## Data analysis and feature prioritisation

All focus group discussions were conducted in English. This was appropriate because participants were required to be sufficiently literate to operate the Lu Dedoo/MASI app, which was available in English, and to participate meaningfully in group discussion. The focus group discussion guide was therefore developed and administered in English and was not translated into local languages. In addition to audio-recording, field notes were taken during each session to document non-verbal expressions, group interactions, and contextual observations that could support interpretation during analysis.

All FGDs were audio-recorded with participants' consent and transcribed verbatim by a trained research assistant familiar with the study protocol. To ensure accuracy, a second trained research assistant and the first author reviewed each transcript against the corresponding audio file and corrected any discrepancies. All transcripts were de-identified before analysis and imported into NVivo® v15 for qualitative data management and coding.

We used thematic analysis following Braun and Clarke's six-step approach [18]. To develop the coding framework, three transcripts were initially coded independently by two researchers. The team then compared codes, discussed areas of overlap and divergence, and agreed on a preliminary coding framework. A combined deductive and inductive approach was used. Deductive codes were derived from the study objectives and the major domains covered in the FGD guide, including app usability, engagement, reminder functions, barriers to use, and suggested modifications. Inductive coding was then applied to identify additional concepts emerging directly from participant narratives that were not fully captured in the initial framework. The coding framework was refined iteratively and then applied across the full dataset. Coding differences were resolved through discussion and consensus. Field notes were used to support interpretation of the transcripts but were not analysed as standalone data sources. The final coding framework comprised three overarching themes, each with related subthemes derived from recurrent codes identified across the dataset (Table 1).

In line with HCD principles, emergent themes were translated into design requirements and feature suggestions. These were then organized and prioritized using the MoSCoW method: Must have: essential features for functionality and user acceptance (e.g., offline access, medication reminders), Should have: important but not critical features (e.g., localized multimedia content), Could have: desirable features to improve user experience (e.g., customizable avatars, gamified reminders), and Won't have (for now): features not feasible in the current development cycle [16].

### App Adaptation and Feature Refinement Process

The Nigerian research team synthesised key thematic findings from the FGDs, with particular attention to participant feedback on usability, content relevance and preferences. These findings were prioritised and translated into specific, actionable recommendations for app modification.

Proposed changes were shared with the app development team through a series of structured virtual meetings. This iterative process ensured that adaptations were both user-informed and technically viable, maintaining the core functionalities of the original platform while enhancing its relevance for AYALHIV in Benue State.

## Results

The first phase of the study involved collecting in-depth feedback from AYALHIV in Benue State, Nigeria, following 1 month of beta testing the MASI app prototype. Thematic analysis of the FGDs highlighted three major themes: User Engagement, App Functionality, and Suggested Modifications for Improvement.

### User engagement

Participants responded positively to the app's engaging features, particularly the interactive and gamified sections. Elements such as quizzes, badges, and games were frequently cited as primary drivers of interest and regular app usage. Several users reported that these elements made the experience enjoyable and rewarding, as one adolescent participant noted:

*"The quiz was interesting. I always checked to see if there was a new one. It felt good when I got answers right." (Male, 20 years old)*

The gamification features, such as badges, enhanced emotional engagement, especially among adolescents aged 15–19. For instance, a female participant shared her experience:

**Table 1. Coding framework used for thematic analysis of focus group discussions.**

| Overarching theme | Subtheme | Contributing codes | Description | Example quotation |
|---|---|---|---|---|
| User engagement | Interactive and gamified features | User experience and engagement; user satisfaction; engagement with app features | Participants described games, quizzes, badges, and other interactive features as enjoyable and motivating, helping sustain interest in app use. | "The games, activities and quiz are fun for me. Earning the badges is interesting." |
| User engagement | Health education and informational value | Content relevance and interest; educational impact of app content | Participants valued educational content, including information on nutrition, sex education, and adherence, and saw this as an important part of the app's usefulness. | "The quiz and articles helps me with tips I need, e.g., nutrition, how to become suppressed." |
| User engagement | Motivation for self-care and treatment engagement | Perceived utility in medication adherence; app impact on behavior; impact on medication adherence behavior | Participants reported that the app increased health awareness, encouraged medication-taking, and strengthened personal commitment to treatment. | "The app helps me take my health seriously." |
| User engagement | Social and peer connection | Social support and interaction; engagement with app features | Participants expressed interest in communication and peer-support features that could foster connection and shared experience. | "The public chat is more preferable to me so you can get to meet new people." |
| App functionality | Ease of navigation and general usability | App reliability; app accessibility; user experience | Participants generally perceived the app as easy to navigate and use after initial familiarisation. | "The app is easy to navigate, it makes it easy to move from one step to the other like you're chatting with someone." |
| App functionality | Navigation, login, and access barriers | App navigation and functionality; barriers to app utilization; challenges in app navigation | Some participants reported difficulties locating the app, entering access codes, or accessing specific features. | "Access code request for the login is giving me difficulty accessing the app." |
| App functionality | Reminder and tracking functions | Reminder effectiveness; perceived effectiveness in medication reminders | Medication reminders, alarms, and tracking functions were perceived as helpful in supporting adherence routines. | "The alarm/notification helps me to take my drugs." |
| App functionality | Internet and technical constraints | Technical challenges; technical support issues | Data costs, poor connectivity, and internet dependence limited consistent access to some app functions. | "I don't always have enough data on my phone." |
| App functionality | Privacy and discretion concerns | Privacy concerns | Some participants expressed concerns about privacy, disclosure, and comfort with interactive features. | "When I was asked to speak to my doctor, I shied away because I am a very secretive person." |
| Suggested modifications for improvement | Offline functionality | Technical challenges; feedback and suggestions; feedback and suggestions for improvement | Participants recommended that core app features should be accessible without internet connectivity. | "The app should be accessed even without data." |
| Suggested modifications for improvement | Customisable and more prominent reminders | Feedback and suggestions; perceived effectiveness in medication reminders | Participants suggested improving reminders through sound, stronger prompts, and greater personalisation. | "The reminders should be a sound (audio) not a silent beep." |
| **Suggested modifications for improvement** | Simpler onboarding and access pathways | App accessibility; app navigation and functionality | Participants recommended streamlining first-time access, login, and app entry processes. | "The link to access the app should be easily sent to your phone not email." |
| **Suggested modifications for improvement** | Enhanced communication and support features | Social support and interaction; feedback and suggestions for improvement | Participants suggested adding or improving chat and expert-support functions to strengthen interaction and support. | "Chat bolt can be added on the Ludedoo app." |

*(Continued)*

**Table 1.** (Continued)

| Overarching theme | Subtheme | Contributing codes | Description | Example quotation |
|---|---|---|---|---|
| **Suggested modifications for improvement** | More locally relevant and youth-friendly content | Content relevance and interest; educational impact of app content | Participants wanted content that was more relatable, engaging, and tailored to their context and age group. | "The app is generally exciting for me. I learn about sex education." |

*"I liked the winning badges. It made me feel like I was doing something good for myself." (Female, 20 years old)*

The self-tracking features, which allowed participants to visualise their adherence behaviours, were also highly valued. These features helped users stay on track with their medication regimen and provided a sense of accountability. One participant explained:

*"I checked my medicine tracker every morning. It helped me stay on time with my drugs." (Female, 21 years old)*

However, participants suggested diversifying the reward system, such as incorporating new levels or customizable avatars, to sustain long-term engagement. Participants' preferences regarding user engagement are summarised in Table 2.

## App functionality

Participants generally found the app easy to navigate (Table 3). One participant shared:

*"I didn't need help to understand where to go or what to do."* (Participant 3, Male, 23 years old)

**Table 2.** Participants' Preferences Regarding User Engagement.

| Category | Description |
|---|---|
| Interactive Features | Participants reported positive experiences with interactive elements such as games, quizzes, and badges, which made the app enjoyable and encouraged regular use. |
| Emotional Engagement | Gamified components, particularly among adolescents aged 15–19, fostered emotional connection and increased engagement with health content. |
| Self-Tracking | Features that allowed users to monitor their medication adherence enhanced self-awareness and a sense of accountability. |
| Motivation | The interactive and reward-based aspects of the app served as external motivators, reinforcing consistent use and supporting ART adherence among AYALHIV. |

**Table 3.** Participants' Preferences for App Functionality and Usabili.

| Category | Description |
|---|---|
| Ease of Navigation | App was generally easy to navigate, especially for those with prior smartphone experience. |
| Reminder Features | Reminders encouraged medication adherence. |
| Internet Data Dependency | App required mobile data. |
| User Support | Suggested tutorial videos to help new users learn app functions. |

However, some participants pointed out challenges related to internet data availability, with one female participant highlighting:

*"It's a good app, but without data, I couldn't watch the videos when my data finished." (Female, 20 years old)*

While the navigation and layout were deemed intuitive, a few participants faced difficulties, particularly with features like setting reminders and accessing the chat function. One participant remarked:

*"I needed help the first time. But after that, it was simple." (Male, 22 years old)*

## Suggested modifications for improvement

Participants suggested several actionable improvements to increase the app's accessibility, relevance, and overall functionality (Table 4).

Participants also requested more frequent updates to multimedia content, including local-language videos, animations, and peer testimonials. These additions were seen as crucial for enhancing both informational content and emotional support. A participant stated:

*"If someone like me talks about taking ARVs in a video, it would encourage others like me. We want to hear from people we can relate to."* (PFemale, 21 years old)

Customizable medication reminders were another frequently mentioned improvement, with users suggesting features like personalized reminder tones, adjustable timing, and motivational messages. One male participant shared:

*"They should include personalized sound or message." (Male, 19 years old)*

Finally, social features such as in-app forums or chat rooms for anonymous peer interaction were proposed to foster a sense of community, reduce stigma, and facilitate the sharing of coping strategies. Additionally, the inclusion of links to local support resources like clinics or helplines was recommended.

Recommendation was made for offline access, especially for features like the medication tracker, daily quizzes, and educational content. One participant expressed:

*"I want to open it anytime, even without data." (Female, 20 years old)*

Table 4. Participants' Suggested Modifications for Improvement.

| Category | Description |
| --- | --- |
| Offline Access | Desire for offline functionality for key features like medication tracker and quizzes. |
| Multimedia Content | Requests for more frequent updates, local-language videos, animations, and testimonials. |
| Customization | Options for personalized reminder tones, timing, and motivational messages. |
| Gamified Reminders | Suggestions to include gamified or musical reminders to enhance engagement. |
| Social Features | Requests for in-app forums or chat rooms for anonymous peer interaction and stigma reduction. |

## Feature Prioritization Using the MoSCoW Method

To translate participant feedback into actionable design requirements, we applied the MoSCoW prioritization framework ("Must have," "Should have," "Could have," "Won't have") [16]. This method provided a structured approach for balancing user preferences with technical feasibility during the adaptation of MASI. This process ensured that the most critical features identified by adolescents and young adults were prioritised for immediate development, while secondary or less feasible suggestions were reserved for later iterations.

## Lu Dedoo app development progress

The Nigerian research team reviewed the MoSCoW prioritisation with the app development team and implemented key modifications. Regular updates of prioritised features were added, including medication tracking and quizzes. Additionally, the team incorporated more multimedia content, including peer-driven games and localised educational materials.

## Discussion

This study describes the use of a modified HCD methodology to adapt a medication adherence app to the local context. This study explored the experiences and preferences surrounding the use of a medication adherence app among AYALHIV in Benue State, Nigeria. Qualitative data from participants were analyzed to inform the contextual adaptation of the MASI app, ensuring its relevance and responsiveness to the unique needs of AYALHIV in this setting. Participants engaged with an early version of the app for one month and shared their feedback in FGDs. Thematic analysis of these discussions informed iterative app modifications that responded directly to user needs and local realities. Overall, participants across all FGDs expressed favorable impressions of the app, describing it as easy to navigate, culturally relevant, and aligned with their preferences for simple, engaging digital tools.

Interactive features such as games, quizzes, badges, forums and medication trackers emerged as key drivers of sustained app engagement. All FGD participants reported willingness to use the app long-term, especially if it could be accessed offline and featured more localized content. Participants suggested design enhancements to improve functionality, accessibility, and emotional resonance. Notably, offline capability, local-language multimedia, customizable reminders, and anonymous peer forums were identified as essential additions.

A few participants reported experiencing initial difficulty using the app's chat and reminder features, underscoring the need for simple user onboarding support. Participants also expressed concerns that Internet data limitations could hinder access to multimedia content. These challenges are consistent with previous studies highlighting the infrastructural barriers to mHealth uptake in low-resource settings [19,20].

Our findings are aligned with a growing body of evidence demonstrating that mHealth interventions are both feasible and effective for supporting HIV care in sub-Saharan Africa. Studies have shown that such apps are particularly appealing to adolescents when they incorporate youth-friendly designs, localized content, and features supporting peer engagement [21,22]. Interactive features and social components have been linked to increased motivation and self-efficacy in ART adherence among young populations [23]. Several studies have similarly applied HCD approaches, using FGDs as a primary method for requirement gathering and app adaptation in mHealth development, reporting positive outcomes in aligning digital tools with user needs [11,23–25].

Our study contributes to this evidence by applying a participatory adaptation process to MASI which was originally developed in South Africa and tailoring it for the Nigerian context. Rather than designing an entirely new app, we followed an efficient adaptation pathway that maintained core functionalities while localizing content, and interface elements. This approach aligns with implementation science principles that call for the reuse and modification of proven tools to accelerate public health impact [26].

In this study, while thematic analysis captured the breadth of adolescent preferences, the application of MoSCoW enabled us to distinguish between features considered essential for usability ("Must have") and those desirable but less critical ("Should have" and "Could have"). This prioritization was crucial for balancing user demands with technical feasibility and resource constraints, ensuring that development efforts focused first on high-impact adaptations such as offline access and medication reminders. Similar to other digital health co-design projects where MoSCoW has been used to integrate youth perspectives into feasible intervention components [27] our study demonstrates the value of combining participatory methods with structured prioritization frameworks.

## Limitations

This study's generalizability is limited by its geographic scope and participant criteria. However, the focused scope of this study could also be a strength rather than a limitation. The purpose of the formative phase was to ensure that the MASI app was meaningfully adapted to Benue State's unique sociocultural and infrastructural context. Recognising that the original South African version of the app may not fully reflect the experiences of Nigerian youth, the localised feedback obtained through these FGDs allowed for precise, context-specific design modifications. While the resulting adaptations may not be directly generalizable to other regions such as Abuja or rural settings, this reflects the very essence of human-centred design, creating digital tools that are responsive to the needs and lived realities of a defined user community.

## Conclusions

This formative study demonstrates the value of human-centered, user-informed design in adapting mHealth solutions for young people living with HIV in Nigeria. The Lu Dedoo Project successfully leveraged youth feedback to guide the contextual adaptation of the MASI app, yielding a more relevant and accessible platform for adherence support. Offline access, peer content, customizable reminders, and simplified onboarding are now being integrated in response to user input.

A pilot trial is underway to assess feasibility, usability, and potential adherence outcomes. If successful, the Lu Dedoo adaptation of MASI could serve as a scalable model for similar digital health interventions across West Africa and other HIV high-burden settings.

## Acknowledgments

The authors wish to express their sincere gratitude to all adolescents and young adults living with HIV who participated in this study and generously shared their time, experiences, and insights during the focus group discussions. Their contributions were instrumental in shaping the Lu Dedoo app's adaptation to reflect the realities and preferences of young people in Benue State. We also acknowledge the dedicated healthcare providers and staff at Northbank General Hospital, Bishop Murray Medical Centre, and Federal Medical Centre, Makurdi, for their invaluable support during participant recruitment and data collection. This is to also appreciate the Desmond Tutu Foundation and APIN Public Health Initiatives staff who were involved in this project.

## Author contributions

**Conceptualization:** Olaposi Joseph Olatoregun, Lisa Hightow-Weidman, Marta Mulawa, Lauren Jennings, Jay Osi Samuels, Prosper Okonkwo, Catherine Orrell.

**Data curation:** Olaposi Joseph Olatoregun.

**Formal analysis:** Olaposi Joseph Olatoregun.

**Funding acquisition:** Lauren Jennings, Catherine Orrell.

**Investigation:** Olaposi Joseph Olatoregun.

**Methodology:** Olaposi Joseph Olatoregun, Lisa Hightow-Weidman, Marta Mulawa, Lauren Jennings, Prosper Okonkwo.

**Project administration:** Olaposi Joseph Olatoregun, Marta Mulawa, Jay Osi Samuels, Prosper Okonkwo, Catherine Orrell.

**Resources:** Olaposi Joseph Olatoregun, Lauren Jennings, Jay Osi Samuels, Catherine Orrell.

**Supervision:** Olaposi Joseph Olatoregun, Lisa Hightow-Weidman, Jay Osi Samuels, Catherine Orrell.

**Validation:** Lisa Hightow-Weidman, Marta Mulawa, Jay Osi Samuels.

**Writing – original draft:** Olaposi Joseph Olatoregun.

**Writing – review & editing:** Olaposi Joseph Olatoregun, Lisa Hightow-Weidman, Marta Mulawa, Lauren Jennings, Jay Osi Samuels, Prosper Okonkwo, Catherine Orrell.

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
