## [Decision Letter · Decision Letter 0]

22 Feb 2026

PONE-D-25-66140Adapting a Medication Adherence App for Adolescents and Young Adults in Makurdi, Benue State: The Lu Dedoo Project"PLOS One

Dear Dr. Olatoregun,

Thank you for submitting your manuscript to PLOS ONE. After careful consideration, we feel that it has merit but does not fully meet PLOS ONE’s publication criteria as it currently stands. Therefore, we invite you to submit a revised version of the manuscript that addresses the points raised during the review process.

**ACADEMIC EDITOR:** The study describes processes of adapting an mHealth software to support adherence among adolescents and young adults living with HIV. This is an important initiative that if well adapted and utilized may improve adherence among the study population. The study design is appropriate, however there are several details lacking in the methodology section. Qualitative study requires very explicit methodology to aid transparency and reproducibility. Please address the following comments in addition to comment provided by the two reviewers below. **Methodology:** The section is lacking several details such as language used in FGD, were the questionnaires in English or local languages, were the translated? Were field notes collected. How was the transcription done from audio to text? How many transcripts were initially coded to develop the coding framework? Please provide the coding framework as figure or describe it explicitly. How was inductive and deductive coding approaches applied? E.t.c.

**Results:** It is unclear about the number of participants. Please provide demographic characteristics of the participants. Also provide details of each group participating in the FGD.

**Acknowledgement:** It was stated that “This work was supported by EDCTP2 (EDCTP TMA2019SFP-2812”, is this the funder? Funding information shouldn’t appear under acknowledgement.

We look forward to receiving your revised manuscript.

Kind regards,

Ibrahim Jahun, MD, MSc, PhD

Academic Editor

PLOS One

Journal Requirements:

[This work was supported by EDCTP2 (EDCTP TMA2019SFP-2812) through Desmond Tutu Foundation, South Africa].

Please provide an amended statement that declares *all* the funding or sources of support (whether external or internal to your organization) received during this study, as detailed online in our guide for authors at http://journals.plos.org/plosone/s/submit-now Please also include the statement “There was no additional external funding received for this study.” in your updated Funding Statement.

3. Please expand the acronym “EDCTP2” (as indicated in your financial disclosure) so that it states the name of your funders in full.

[This is to also appreciate the Desmond Tutu Foundation and APIN Public Health Initiatives staff who were involved in this project. This work was supported by EDCTP2 (EDCTP TMA2019SFP-2812)]

[This work was supported by EDCTP2 (EDCTP TMA2019SFP-2812) through Desmond Tutu Foundation, South Africa]

5. Please note that your Data Availability Statement is currently missing the repository name and/or the DOI/accession number of each dataset. If your manuscript is accepted for publication, you will be asked to provide these details on a very short timeline. We therefore suggest that you provide this information now, though we will not hold up the peer review process if you are unable.

7. We note that Figures 1, 2, and 3 in your submission contain copyrighted images. All PLOS content is published under the Creative Commons Attribution License (CC BY 4.0), which means that the manuscript, images, and Supporting Information files will be freely available online, and any third party is permitted to access, download, copy, distribute, and use these materials in any way, even commercially, with proper attribution. For more information, see our copyright guidelines: http://journals.plos.org/plosone/s/licenses-and-copyright.

1. You may seek permission from the original copyright holder of Figures 1, 2, and 3 to publish the content specifically under the CC BY 4.0 license.

8. We note you have included a table to which you do not refer in the text of your manuscript. Please ensure that you refer to Tables 1, 2, and 3 in your text; if accepted, production will need this reference to link the reader to the Tables.

9. We note that your manuscript consists of interview transcripts. Can you please confirm that all participants gave consent for interview transcript to be published?

If they DID provide consent for these transcripts to be published, please also confirm that the transcripts do not contain any potentially identifying information (or let us know if the participants consented to having their personal details published and made publicly available). We consider the following details to be identifying information:

- Names, nicknames, and initials

- Age more specific than round numbers

- GPS coordinates, physical addresses, IP addresses, email addresses

- Information in small sample sizes (e.g. 40 students from X class in X year at X university)

- Specific dates (e.g. visit dates, interview dates)

- ID numbers

Or, if the participants DID NOT provide consent for these transcripts to be published:

- Provide a de-identified version of the data or excerpts of interview responses

- Provide information regarding how these transcripts can be accessed by researchers who meet the criteria for access to confidential data, including:

a) the grounds for restriction

b) the name of the ethics committee, Institutional Review Board, or third-party organization that is imposing sharing restrictions on the data

c) a non-author, institutional point of contact that is able to field data access queries, in the interest of maintaining long-term data accessibility.

d) Any relevant data set names, URLs, DOIs, etc. that an independent researcher would need in order to request your minimal data set.

For further information on sharing data that contains sensitive participant information, please see: https://journals.plos.org/plosone/s/data-availability#loc-human-research-participant-data-and-other-sensitive-data

If there are ethical, legal, or third-party restrictions upon your dataset, you must provide all of the following details (https://journals.plos.org/plosone/s/data-availability#loc-acceptable-data-access-restrictions):

1. A complete description of the dataset

2. The nature of the restrictions upon the data (ethical, legal, or owned by a third party) and the reasoning behind them

3. The full name of the body imposing the restrictions upon your dataset (ethics committee, institution, data access committee, etc.)

4. If the data are owned by a third party, confirmation of whether the authors received any special privileges in accessing the data that other researchers would not have

5. Direct, non-author contact information (preferably email) for the body imposing the restrictions upon the data, to which data access requests can be sent

Reviewers' comments:

Reviewer's Responses to Questions

**Comments to the Author**

1. Is the manuscript technically sound, and do the data support the conclusions?

Reviewer #1: Yes

Reviewer #2: Yes

2. Has the statistical analysis been performed appropriately and rigorously? 

Reviewer #1: Yes

Reviewer #2: Yes

3. Have the authors made all data underlying the findings in their manuscript fully available?

Reviewer #1: Yes

Reviewer #2: Yes

4. Is the manuscript presented in an intelligible fashion and written in standard English?

Reviewer #1: Yes

Reviewer #2: Yes

5. Review Comments to the Author

Reviewer #1: The author should clarify the following:

1. Regarding participants recruitment refer to line 119 provider referral and line 149 consecutive recruitment

2. Regarding number of study sites refer to line 119 study was conducted at three HIV treatment clinics versus line 150 two study sites

3. Line 155 revise transport stipend of USD 10 equivalent in local currency

Reviewer #2: The author should clearly define what MASI stands for at first mention.

Participants should be selected based on documented non-adherence in order to properly validate the effectiveness of the app.

Inferential statistical analysis should be used to compare adherence levels, using viral load results before and after the intervention.

The study should follow newly enrolled clients for seven months and stable clients for one year to allow for meaningful viral load assessment.

A focus group discussion (FGD) should be incorporated alongside the quantitative analysis to capture participants’ perspectives and experiences.

Overall, it is not possible to conclude that the app improves adherence among AYALHIV without supporting clinical data to provide objective evidence.

6. PLOS authors have the option to publish the peer review history of their article (what does this mean?). If published, this will include your full peer review and any attached files.

Reviewer #1: No

Reviewer #2: **Yes:** Agboola Samson

---

## [Author Response · Author response to Decision Letter 1]

7 Apr 2026

Dear Dr. Jahun and Reviewers,

Thank you for the careful review of our manuscript and for the constructive comments. We appreciate the opportunity to revise the manuscript. We have carefully All changes have been incorporated into both the tracked and clean versions of the manuscript.

Below, we provide a point-by-point response.

Response to the Academic Editor

Comment 1: Qualitative methodology requires more detail

“The section is lacking several details such as language used in FGD, were the questionnaires in English or local languages, were the translated? Were field notes collected. How was the transcription done from audio to text? How many transcripts were initially coded to develop the coding framework? Please provide the coding framework as figure or describe it explicitly. How was inductive and deductive coding approaches applied? E.t.c.”

Response:

Thank you for this important comment. We have substantially revised the qualitative methodology section to improve transparency and reproducibility. Specifically, we have now clarified that all FGDs were conducted in English; the discussion guide was developed and administered in English and was not translated; field notes were collected during each discussion; all FGDs were audio-recorded and transcribed verbatim by a trained research assistant; transcript accuracy was checked against the audio files by a second trained research assistant together with the first author; and three transcripts were initially coded to develop the preliminary coding framework. We have also explicitly described the combined deductive and inductive coding approach and added a coding framework table.

Comment 2: Participant numbers and group details are unclear

“It is unclear about the number of participants. Please provide demographic characteristics of the participants. Also provide details of each group participating in the FGD.”

Response:

Thank you. We have revised the manuscript to clearly report the final achieved FGD sample and group composition. The revised manuscript now states that a total of 22 participants took part in three FGDs: one mixed-gender adolescent group aged 15–19 years (n = 6), one male young adult group aged 20–24 years (n = 8), and one female young adult group aged 20–24 years (n = 8).

Comment 3: Funding information should not appear in Acknowledgments

“It was stated that ‘This work was supported by EDCTP2 (EDCTP TMA2019SFP-2812’, is this the funder? Funding information shouldn’t appear under acknowledgement.”

Response:

Thank you. We have removed all funding-related text from the Acknowledgements section. Funding information is now presented only in the Funding Statement, in accordance with PLOS ONE requirements.

Response to Reviewer 1

Comment 1: Recruitment wording is inconsistent

Comment:

“Regarding participants recruitment refer to line 119 provider referral and line 149 consecutive recruitment”

Response:

Thank you for noting this inconsistency. We have revised the manuscript to harmonise the description of recruitment. Participants were identified during routine clinic visits with support from clinic staff and were then enrolled consecutively from among those who met the eligibility criteria and agreed to participate.

Comment 2: Number of study sites is inconsistent

“Regarding number of study sites refer to line 119 study was conducted at three HIV treatment clinics versus line 150 two study sites”

Response:

Thank you. We have corrected this inconsistency in the revised manuscript. The study setting and participant recruitment sections now consistently reflect the actual qualitative study sites used for the FGDs. The study was done in 3 facilities.

Comment 3: Transport stipend wording

“Line 155 revise transport stipend of USD 10 equivalent in local currency”

Response:

Thank you. We have revised this wording in the manuscript to improve clarity and local relevance. The revised text now states the amount received in Naira

Response to Reviewer 2

Comment 1

“The author should clearly define what MASI stands for at first mention.”

Response:

Thank you. We have revised the manuscript to define MASI in full at first mention as Masakhane Siphucule Impilo Yethu.

Comment 2: Participants should be selected based on documented non-adherence to validate effectiveness

Comment:

“Participants should be selected based on documented non-adherence in order to properly validate the effectiveness of the app.”

Response:

Thank you for this thoughtful comment. We respectfully clarify that the present manuscript reports the formative qualitative adaptation phase of the Lu Dedoo project rather than the effectiveness phase. The purpose of this manuscript is to describe how the MASI app was contextually adapted for adolescents and young adults living with HIV in Benue State using user-centered qualitative input. It is therefore focused on usability, acceptability, and adaptation needs rather than validation of clinical effectiveness.

Comment 3: Inferential statistical analysis should compare adherence levels using viral load results before and after intervention

Comment:

“Inferential statistical analysis should be used to compare adherence levels, using viral load results before and after the intervention.”

Response:

Thank you. We agree that such analyses are important for evaluating effectiveness. However, these analyses fall within the subsequent pilot randomized component of the larger study and are outside the scope of the present manuscript, which is focused on the formative qualitative adaptation phase.

Comment 4: Follow newly enrolled clients for seven months and stable clients for one year for viral load assessment

Comment:

“The study should follow newly enrolled clients for seven months and stable clients for one year to allow for meaningful viral load assessment.

Response:

Thank you for this suggestion. We agree that viral load follow-up duration is an important consideration for the later effectiveness study. However, this recommendation pertains to the pilot trial phase and not to the current qualitative adaptation manuscript. We have therefore not introduced viral load outcome analysis into this paper.

Comment 5: A focus group discussion should be incorporated alongside quantitative analysis

Comment:

“A focus group discussion (FGD) should be incorporated alongside the quantitative analysis to capture participants’ perspectives and experiences.”

Response:

Thank you. We agree with this point, and indeed, the current manuscript is centred on the FGD component of the study. The present paper specifically reports on the formative focus group phase conducted prior to the pilot trial to capture participants’ perspectives, app-use experiences, and needs for contextual adaptation.

Comment 6: It is not possible to conclude that the app improves adherence without supporting clinical data

“Overall, it is not possible to conclude that the app improves adherence among AYALHIV without supporting clinical data to provide objective evidence.”

Response:

Thank you. the current manuscript is centred on the FGD component of the study. The present paper specifically reports on the formative focus group phase conducted prior to the pilot trial to capture participants’ perspectives, app-use experiences, and needs for contextual adaptation of the app.

Response to Journal Requirements

Requirement 1: PLOS ONE style requirements

Response:

We have revised the manuscript to align with PLOS ONE formatting and style requirements and will submit the files using the required naming convention.

Requirement 2: Funding statement must include all support and state no additional external funding

Response:

We have prepared an amended Funding Statement for the cover letter and submission system:

Funding Statement:

“This work was supported by the European & Developing Countries Clinical Trials Partnership 2 (EDCTP2) programme under grant number EDCTP TMA2019SFP-2812, through the Desmond Tutu Health Foundation, South Africa. There was no additional external funding received for this study.”

Requirement 3: Expand EDCTP2 in full

Response:

We have expanded EDCTP2 in full in the Funding Statement and cover letter as European & Developing Countries Clinical Trials Partnership 2 (EDCTP2).

Requirement 4: Remove funding text from the manuscript body/Acknowledgments

Response:

We have removed funding-related text from the Acknowledgments section and retained funding information only in the Funding Statement. The editor letter specifically noted the prior Acknowledgments wording.

Requirement 5: Data Availability Statement should include repository/access details

Response:

We have revised the Data Availability Statement to provide the repository name and access details for the de-identified dataset.

Requirement 6: Ethics statement should only appear in the Methods section

Response:

We have moved the ethics statement so that it appears only in the Methods section and removed it from any other section of the manuscript.

Requirement 7: Copyrighted figures

Response:

We have deleted Figures 1, 2, and 3

Requirement 8: Tables 1, 2, and 3 must be cited in text

Response:

We have revised the Results section to ensure that Tables 1, 2, and 3 are explicitly referred to in the text.

Requirement 9: Clarify consent for interview transcript publication and de-identification

Response:

We confirm that the manuscript includes only de-identified excerpts from focus group discussions and not full raw transcripts. All quotations were reviewed to ensure that they do not contain names, initials, dates, addresses, contact details, ID numbers, or other direct identifiers.

---

## [Editor Report · Decision Letter 1]

13 May 2026

Adapting a Medication Adherence App for Adolescents and Young Adults in Makurdi, Benue State: The Lu Dedoo Project"

PONE-D-25-66140R1

Dear Dr. Olatoregun,

We haven't heard from both reviewers who reviewed the manuscript after reminders. I have however reviewed the responses provided to my comments and to both reviewers' comments and I am convinced that the responses provided are adequate. The major concern about this paper was the qualitative methodology as provided in my comments, and you have adequately responded to the comments.

At this point, we’re therefore pleased to inform you that your manuscript has been judged scientifically suitable for publication and will be formally accepted for publication once it meets all outstanding technical requirements.

Kind regards,

Ibrahim Jahun, MD, MSC, PhD

Academic Editor

PLOS One
---

## [Editor Report · Acceptance letter]

PONE-D-25-66140R1

PLOS One

Dear Dr. Olatoregun,

I'm pleased to inform you that your manuscript has been deemed suitable for publication in PLOS One. Congratulations! Your manuscript is now being handed over to our production team.

Kind regards,

on behalf of

Dr. Ibrahim Jahun

Academic Editor

PLOS One